# Biomaterials in Gastroenterology: A Critical Overview

**DOI:** 10.3390/medicina55110734

**Published:** 2019-11-12

**Authors:** Adrian Goldis, Ramona Goldis, Traian V. Chirila

**Affiliations:** 1Faculty of Medicine, Victor Babes University of Medicine and Pharmacy, 300041 Timisoara, Romania; 2Algomed Policlinic, 300002 Timisoara, Romania; amalia_goldis@yahoo.com; 3Queensland Eye Institute, South Brisbane, QL 4101, Australia; traian.Chirila@qei.org.au; 4Science & Engineering Faculty, Queensland University of Technology, Brisbane, QL 4000, Australia; 5Faculty of Medicine, University of Queensland, Herston, QL 4029, Australia; 6Australian Institute for Bioengineering and Nanotechnology, University of Queensland, St Lucia, 4072 QL, Australia; 7Faculty of Science, University of Western Australia, Crawley, WA 6009, Australia; 8University of Medicine, Pharmacy, Sciences and Technology, 540139 Targu Mures, Romania

**Keywords:** biomaterials, biocompatibility, gastrointestinal tract, endoscopic devices, hemostasis, surgical adhesives, stents, bariatric devices

## Abstract

In spite of the large diversity of diagnostic and interventional devices associated with gastrointestinal endoscopic procedures, there is little information on the impact of the biomaterials (metals, polymers) contained in these devices upon body tissues and, indirectly, upon the treatment outcomes. Other biomaterials for gastroenterology, such as adhesives and certain hemostatic agents, have been investigated to a greater extent, but the information is fragmentary. Much of this situation is due to the paucity of details disclosed by the manufacturers of the devices. Moreover, for most of the applications in the gastrointestinal (GI) tract, there are no studies available on the biocompatibility of the device materials when in intimate contact with mucosae and other components of the GI tract. We have summarized the current situation with a focus on aspects of biomaterials and biocompatibility related to the device materials and other agents, with an emphasis on the GI endoscopic procedures. Procedures and devices used for the control of bleeding, for polypectomy, in bariatrics, and for stenting are discussed, particularly dwelling upon the biomaterial-related features of each application. There are indications that research is progressing steadily in this field, and the establishment of the subdiscipline of “gastroenterologic biomaterials” is not merely a remote projection. Upon the completion of this article, the gastroenterologist should be able to understand the nature of biomaterials and to achieve a suitable and beneficial perception of their significance in gastroenterology. Likewise, the biomaterialist should become aware of the specific tasks that the biomaterials must fulfil when placed within the GI tract, and regard such applications as both a challenge and an incentive for progressing the research in this field.

## 1. Introduction

Biomaterials are materials, other than drugs, developed to be used in medicine. They are the essential components of the rapidly evolving fields of applied tissue engineering and regenerative medicine. In their day-to-day practice, clinicians and surgeons are making extensive use of biomaterials, likely without being aware of this term as such. Indeed, the medical devices for diagnosis and intervention, including surgical implants, are all made from materials that qualify as “biomaterials” as we define them today. Although the term has a variety of definitions, all are conveying the same general idea that a biomaterial is a natural or a man-made material that was developed to be inserted into the human body with a precise diagnostic or therapeutic aim. An earlier definition [1] specifies that a biomaterial is a material that must interface with biological systems with an aim “to evaluate, treat, augment or replace any tissue, organ or function of the body”. According to the most recent definition [2], “a biomaterial is a substance that has been engineered to take a form which, alone or as a part of a complex system, is used to direct, by control of interactions with components of living systems, the course of any therapeutic or diagnostic procedure”. Certain confusing ambiguities related to definitions of a biomaterial have been discussed recently [2,3], concluding that the term should be strictly employed for any material used to make devices that impact directly on human health, regardless of its origin.

The investigators involved in the study and development of biomaterials include mainly non-medical specialists, such as materials/polymer scientists, chemical engineers, biologists, biochemists, and immunologists. Medical doctors have been implicated in the study of biomaterials to a rather limited extent for various reasons. Their main priority is obviously the treatment and care of patients, an activity associated with a very demanding workload in clinics and hospitals that allows little time for personal involvement in the study of other disciplines. 

### 1.1. The Concept of Biocompatibility

An ideal biomaterial should be biologically acceptable to the host living cells and tissues in order to fulfil its intended role in the diagnosis of pathologic conditions, replacement therapies, or regeneration of tissues. In other words, once its intended biofunctionality has been established, it becomes mandatory that a biomaterial candidate shall also display biocompatibility. The latter term appears self-explanatory, but in fact is rather a complex concept that has been used for decades without a complete understanding [4]. A material is biocompatible when is able to perform according to the expected purpose in a specific application in human body and deliver an appropriate response; clearly, biocompatibility shall be defined only in relation to a specific application. Biocompatibility ultimately involves the interaction of living tissues with the surface of the biomaterial components present in devices or implants. The quality of such interaction, as a fundamental criterion for biocompatibility, is commonly gauged through the ability of the biomaterial surface to promote and maintain the attachment, spreading, proliferation and differentiation of living cells. However, this is not sufficient: the simple presence of leachable toxic substances in the biomaterial makes it non-biocompatible, regardless of any other more subtle aspects of the surface-cell interaction. Indeed, this is a common situation that especially involves the synthetic polymeric biomaterials, where diffusion of toxic contaminants, such as unreacted monomers, traces of inhibitors, initiators and other additives, into the cells’ environment can cause damage or death of cells, foreign body reaction, and eventual rejection of the implanted material. Further biocompatibility assays become meaningless for such materials, which shall no longer be qualified as biomaterials and/or used in this capacity. Conversely, the lack of residual toxic contaminants in a material is not a sufficient justification to assert its biocompatibility and consider it a true biomaterial. For instance, certain polymers may display inherent cytotoxicity as a consequence of structural factors at a molecular level.

### 1.2. Biomaterials and the Gastrointestinal Tract

When it comes to the application of biomaterials in gastroenterology, some particular aspects are to be discussed. First, one can notice that there is little attention dedicated to gastrointestinal biomaterials within the major textbooks or reviews published in the field of biomaterials science. Second, reviews or reports on the clinical performance of various therapeutic devices in gastrointestinal applications do not discuss aspects related to materials science; in fact, information on the nature of materials in these publications is scant, if any. It becomes quite difficult for a biomaterials expert to make a meaningful appraisal of the biomaterials used in the gastrointestinal tract when their chemical nature is not disclosed. Our overview aims at contributing to amend this state of affairs. Further, at a first glance, the use of biomaterials in the gastrointestinal (henceforth GI) tract seems to involve exclusively the contact between materials and epithelia covering the mucosal surfaces that are lining the tract. This situation elicits a question that by now has become classic: can the materials that contact only epithelial layers be regarded as biomaterials? According to most investigators, the answer is affirmative. To illustrate the point, neither the glass in the lenses of spectacles, nor the polymers and metals in the design of stethoscopes or artificial limbs can be regarded as biomaterials because their contact with parts of our body is episodic, superficial, or lacking altogether, and does not involve contact with body fluids or penetration into the body. In contradistinction, the materials employed to make contact lenses or stents are to be considered true biomaterials due to the direct contact with epithelial/mucosal surfaces during their applications. We will discuss the materials used currently to fabricate implantable devices for gastroenterologic therapeutic purposes, but only those placed within the components of GI tract, therefore contacting the mucosal surfaces.

Mucosae in the GI tract contain three principal components: the epithelium (multiple layers in the esophageal epithelium but a single layer in intestines and stomach), lamina propria (subepithelial connective tissue), and the smooth muscle layer. The mucosae are equipped with defense strategies against pathogens, which include mechanical (the preepithelial mucus, the cells as such, and ciliary and desquamation activities), chemical (antimicrobial proteins, cytokines, pattern recognition molecules), and cellular (phagocytes, dendritic cells, mast cells, lymphocytes etc.) components [5]. In certain situations these defense strategies are overwhelmed, and medical treatment is required. Inspired by the mechanisms underlying the attack by pathogens at the GI mucosal level, a number of scenarios can be imagined leading to damage caused by non-biocompatible foreign materials such as polymers. For instance, the release of a toxin from the material results in injury to epithelium and loss of the epithelial barrier. It may cause a prolonged opening of the chloride channels that are instrumental in maintaining balance between secreted and absorbed water in the small intestine. It can also lead to the recruitment of neutrophils and promotion of infiltrates that can further trigger necrosis of the upper mucosa. If reactive oxygen species become involved, the epithelial cells will be damaged or killed, and the immature cells replacing them are deficient in the enzymes and transporters that are essential for absorption of water and nutrients. All these events can cause severe complications to the patient.

Many therapeutic procedures for the GI tract, such as hemostasis of varices or alleviation of luminal obstructions, rely on endoscopy, and our overview will dwell on the biomaterials related to endoscopic procedures performed by gastroenterologists. While the primary role of an endoscope is to function as a diagnostic tool, in the present article we will only discuss the materials involved in the fabrication of the endoscopes that are used as a therapeutic tool. The insertion tube and the distal tip of endoscopes contains metals, polymers and glass (as lenses). Some of these materials come in contact with mucosal surfaces, so they have to be biocompatible and uncontaminated, therefore qualifying as proper biomaterials. The polymers involved in the manufacture of the insertion tube include silicones, polyurethanes, polyesters and others, either as such or as blends, composites, or multilayers. In addition to displaying biocompatibility, these materials must also be resistant to chemical attack and to heat, the latter being a requirement due to the mandatory sterilization procedures. The insertion tube commonly consists of two layered polymers, of which the top coating polymer is particularly relevant to the functionality of an endoscope; the presence of buckles or cracks on the outer surface of the tube prohibits its use. The natural orifices in our body that connect the hollow organs with the exterior are safe sites for the insertion of the endoscope and do not raise biocompatibility issues. Insertion of the endoscope through small incisions into cavities at safe locations may raise such concerns, but the materials currently employed for the manufacture of endoscopes are biocompatible and no significant tissue reaction is expected. Severe complications of gastrointestinal endoscopy can be caused either accidentally or due to improper maintenance or cleaning/sterilization of the instrument, and they may outweigh the benefits of procedure. Such complications include perforation of the gut wall, bleeding, infection, and patient’s untoward reaction to sedation. Biocompatibility can become indeed an issue in the case of a perforation; however, the damage to the tissue is of a much greater concern than the foreign-body response of the tissue to the polymeric or metallic components of the distal tip and insertion tube. The use of capsules, developed over the past two decades [6] to enable the visualization of the small bowel structures, posits indeed more stringent biocompatibility issues. These miniature video cameras, encapsulated in an enclosure of polycarbonate, are to be swallowed by the patient. Although they are supposedly excreted in feces within 24 to 48 hours, there are a large number of reported cases where the retention of the capsules lasted much longer [7]. Due to possible intestinal obstruction or perforation as complications, the capsules are usually removed by either surgery, endoscopy, or pharmaceutical manipulation. However, to date there has been no reported complication due to deterioration of the outer polymer capsule even after a very long retention, which bears testimony to the remarkable resistance of the device’s polycarbonate outer layer.

### 1.3. Current Developments

Biomaterials are essential tools in the development of tissue engineering (henceforth TE), a discipline that aims at creating functional substitutes able to maintain, restore, or improve tissue functions that were compromised by trauma, disease, or aging. Importantly, the biomaterials used in TE must be able to interact and fully integrate with host cells and tissues, and therefore they have to be modified chemically and/or morphologically, i.e., engineered for specific purposes. TE extends beyond the concept of implanted prostheses or devices, and is aimed at generating new tissue for the reconstruction of parts of human body by enticing the cells to achieve this in circumstances that for them are not normal [8].

Tissue engineering is regarded as a major component of the regenerative medicine (henceforth RM), a field that currently enjoys many definitions reflecting the multitude of opinions regarding the aim of this discipline. In short, RM “replaces or regenerates human cells, tissue or organs, to restore or establish normal function” [9]. Traditional transplantation and replacement therapies do not belong to RM, as “replacement” is different from “regeneration”. Furthermore, fundamental to RM is the distinction between “repair” and “regeneration” [10]. The spontaneous repair of a wound by formation of scar tissue does not restore the original structural integrity and function; this can be achieved only by a more advanced healing process that shall involve the re-synthesis of missing tissue or organ and the restoration of their normal functionality. It is the task of RM to find the methods of inducing regeneration.

The GI tract is an inordinately complex part of our body, both anatomically and physiologically. To repair its damage caused by cancer, inflammatory diseases or incontinence requires strategies more effective than those based on the current therapeutic armamentarium. Both TE and RM made spectacular inroads in gastroenterology, to such extent that the term “regenerative gastroenterology” has been already coined [11]. The progress in this field will be presented in a future overview.

## 2. Obturation Procedures for Esophageal and Gastric Varices

Fundamentally, the esogastric varices, commonly differentiated into esophageal varices (EVs) and gastric varices (GVs), are dilated submucosal veins of the GI tract. They are frequently the result of portal hypertension, a progressive complication of chronic liver diseases at their end stages, mainly cirrhosis. The rupture of varices, succeeded by inevitable bleeding, constitutes a major medical emergency and is the prevalent cause of death in cirrhotic patients. Due to a higher blood flow, GVs are more severe than EVs and lead to a higher mortality rate, but they occur less frequently.

Over the past five decades, gastroenterologists have made tremendous efforts to develop and apply treatment procedures for the gastric varices, as primary or secondary prophylactic therapies [12,13,14,15,16,17,18]. The endoscopic procedures are the preferred therapeutic approaches for treating EVs and GVs, although the choice of the procedure relies ultimately on an accurate identification of the varix type. According to a widely accepted classification based on their location [14,19,20], sanctioned by the Baveno consensus working group, there are two types of EVs designated as GOV-1 (localized in the lesser curvature) and GOV-2 (in the fundus and extending to the greater curvature), and two types of isolated GVs designated as IGV-1 (in the gastric fundus) and IGV-2 (in other parts of the stomach).

From the current procedures available for treatment of esogastric varices, we will discuss only those involving application of biomaterials, can be implemented endoscopically, and treat tissues and organs located within the upper or lower GI tract. Such procedures include: balloon tamponade; variceal band ligation, self-expanding esophageal stents; hemostatic agents (hemospray); and obturation with cyanoacrylate glue. 

### 2.1. Balloon Tamponade for Variceal Bleeding

In principle, this non-endoscopic procedure consists of direct compression exerted on the varix though an inflated balloon, combined with mechanical traction exerted from outside through a suspended weight to induce pressure on the cardia or fundus, with the aim of interrupting the submucosal venous blood flow. It is one of the oldest non-surgical approach to obturate varices and is regarded rather as an emergency or temporary procedure bridging to other therapies, mainly when hemostatic techniques failed or when endoscopy is not feasible or readily available. Inflated balloons were first reported in 1947 by Rowntree et al. [21], who used them in two cirrhotic patients. The device, consisting of a latex rubber balloon attached at the distal end of a tube, stopped the bleeding after being kept in place for 4 days and, respectively, 40 h. Further improvements [22,23,24,25,26,27] resulted in a number of balloon types to be used as tamponade for variceal bleeding. Usually the inflated balloon and traction are maintained in place for 24–48 h, and afterwards the deflated balloon is kept in place for an additional 24 h or so. The complications related to the procedure [28,29,30,31,32,33,34] and the general performance of the balloon devices have been episodically reviewed [35,36].

Generally, the balloon tamponade is regarded as a technique that may be effective in controlling acute variceal bleeding, but offers only transient hemostasis and is associated with a high rate of complications. In fact, the balloon therapy for variceal bleeding is considered nowadays as the last-preferred method of treatment. 

There have been many evolving types of balloons, also termed as “tubes”, used in the management of bleeding EVs or GVs, the most common now being the Sengstaken-Blakemore, Linton (or Linton-Nachlas) and Minnesota models. In time, the manufactured structure of these devices became increasingly sophisticated for the benefit of both doctors and patients, leading to a multitude of different biomaterials being included into a single device. Unfortunately, the manufacturers disclose little about the materials, likely because of market competition. On the other hand, to find details about balloon materials in the gastroenterological literature is rather out of the question, although the pioneers of the procedure actually mentioned latex rubber as the material of the balloon [21,23]. However, commercial information that is available online indicates that the biomaterials employed in the manufacture of medical balloons and their attached tubes include polyamides (e.g., Nylon^®^, Pebax^®^), polyethylene-terephthalate (PET), polyurethanes, silicone rubber, latex rubber, polyolefins (e.g., polyethylene, polypropylene) and their copolymers (e.g., EVA), polytetrafluoroethylene (i.e., Teflon^®^), polyimides, polyvinylidene-fluoride and polyketones. To lower the friction between the tube and the mucosal layers, other polymers are used to coat the external surface of the tubes; the materials for coating are usually hydrophobic polymers (polyolefins, acrylics, polyurethanes, etc.) mixed with polyethylene-glycols. As such, the balloons can be compliant (distensible, stretchable), non-compliant, and semi-compliant, and the polymers can be selected or modified (by blending, chemical grafting, etc.) to match any of those types. The balloons used for varix obturation would likely belong to the non- or semi-compliant categories.

The complications associated with the balloon tamponade are both numerous and difficult to manage [28,29,30,31,32,33,34,35,36]. The major and potentially lethal complications include esophageal rupture, failure to control bleeding, pulmonary aspiration and pneumonia, airway obstruction, and chest infection. Minor complications are associated with nasal damage (excoriation, bleeding, injury, or necrosis) and chest pain. Neither the nature of materials in the device nor biocompatibility issues have been mentioned in these reports, which is rather disquieting considering that most of materials in balloons and tubes could have contained traces of chemical additives such as initiators, crosslinking agents and plasticizers, all toxic to living tissues. Therefore, we cannot draw a valid conclusion on the effect of balloon biomaterials on the reported clinical outcomes of the procedure. It may be that the residence of balloons (with appendages) in contact with GI mucosae was too short to elicit a significant host tissue response to foreign materials, but complications such as chest infection or nasal necrosis could be associated with biomaterial issues. It is expected, however, that the manufacturers of tamponade balloons fulfilled their duty to evaluate in their research and development (R&D) laboratories the response of host GI tissues to the balloon materials. Indeed, all biomaterials mentioned above display biocompatibility in those medical applications for which they had been developed [37], but such information regarding the tamponade balloons for varices is missing in the gastroenterological literature.

### 2.2. Endoscopic Variceal Band Ligation

The endoscopic variceal band ligation, known alternatively as endoscopic banding ligation (EBL), endoscopic variceal ligation (EVL), or variceal band ligation (VBL), is an established procedure for the control of variceal bleeding and rebleeding. Following extensive animal experimentation [38,39], Van Stiegmann and colleagues further developed and applied for the first time the procedure to human patients [40,41,42]. They also provided a detailed description of the device, and discussed the mechanism of varix obliteration. In principle, mechanical strangulation with an elastic band of the variceal tissue leads to ischemic necrosis with acute inflammation and shallow ulcer formation. The healing process leads to the obturation of venous channels by dense scar formation, and the dead tissue sloughs away. The device for EVL was a modified endoscopic overtube, consisting of housing cylinder, banding cylinder (with a notch), conical loading device (attached to the banding cylinder), trip wire with flanges, and rubber O-ring band (placed onto the loading device). The originators of EVL have applied the procedure to over 100 patients including more than 400 sessions, and concluded that it is a safe and effective treatment for the esophageal varices of 1 cm or smaller [40,41,42]. No major complications have been reported except for episodes of recurrent bleeding, esophageal strictures, and pneumonia.

EVL has become an effective procedure for both primary and secondary prophylaxis in human patients with EVs not larger than 2 cm, leading to less complications than the therapies with sclerotizing agents or β-blockers [13,14,16,43,44,45,46,47,48,49]. However, a combination of EVL with pharmaceutical approaches has been advocated. The isolated GVs, much less frequent but larger in size and more problematic than EVs, have been seldom treated with EVL [13,14,17,47,50,51]. Inspired from procedures developed for the endoscopic resection of elevated lesions, such as large pedunculated polyps, detachable snares (endoloops) have been used (alone [52] or associated with sclerotherapy [53]) for controlling bleeding from GVs. In subsequent trials [54,55], successful hemostasis has been reported using detachable endoloops made of stainless steel or Nylon^®^ (a polyamide). After their detachment, the snares are left on site and eventually they fall off and are spontaneously eliminated after passing through the tract. However, the endoloops are currently used almost exclusively for polypectomy. 

The adverse effects of the EVL have been variably reported, and in general they are manageable. Such complications include: retrosternal pain, fever, transient dysphagia, esophageal ulcers or strictures, iatrogenic bleeding, transient bacteremia, peritonitis. In only one case [56], transient bacteremia has been associated with possible injury caused to the oropharyngeal mucosa by the endoscope or the elastic band. To our knowledge, none of the other complications described in the literature have been explicitly correlated to the biomaterials in the ligation devices and loops. 

In the early EVL devices, the predominant material coming in contact with the mucosae was stainless steel, an alloy containing iron, chromium, nickel and carbon. Metallic biomaterials have been used as implant materials for decades [57,58], although some alloys like those containing nickel can trigger allergic reactions. In EVL, as the contact between the metallic components and tissue is relatively short; therefore, chemically induced effects are unlikely. However, the environment of the GI tract is mechanically dynamic and therefore there must be a compliance match between the inserted metallic biomaterials and host tissues. This will allow for the biomaterial to sustain and recover from deformations that otherwise can lead to mechanical irritation of the surrounding tissues and further complications (inflammation, scar formation, etc.). Therefore, it is a matter of concern that mechanical irritation due to contact between tissues and rigid metals has not been investigated to a larger extent as a potential cause for post-ligation complications. In the current EVL devices, the stainless steel in the cylindrical loading components has been replaced with polymers, and the endoscopic overtube has been removed. Multi-band ligators are used almost exclusively nowadays, and manufacturers such as Boston Scientific Corporation, Cook Endoscopy, Olympus America Inc. or Scandimed International can provide ligators with up to 10 pre-loaded ligating bands. Latex rubber has been generally replaced by non-latex materials, and anti-slip bands have become a common feature. Most of these improvements are based on proprietary concepts and consequently the nature of materials or their structural modification have not been disclosed, which makes redundant further discussion. The fact that the ligating bands or the detachable snares remain inside the tract after procedure and they are made of materials that are not biodegradable in short term should raise some biocompatibility concerns.

### 2.3. Self-Expanding Esophageal Stents for Variceal Bleeding

Endoscopic stenting has been primarily developed as a palliative treatment for the patients with cancerous obstructions located within the GI. Higher rates of complications induced by non-expandable or expandable stents made from synthetic polymers (silicone rubber, latex rubber, polyethylene, polyurethanes, Teflon^®^, etc.), such as migration, perforation, or occlusion, have prompted the introduction of self-expanding (or “self-expandable”) metal stents (SEMSs), which nowadays are preferentially used for the palliation therapy of inoperable gastrointestinal malignancies [59,60,61,62,63]. Palliative stenting will be discussed in a later section. We should mention here that esophageal stenting is extensively used to dilate permanently esophageal stenosis, either of a benign nature or a result of malignancies, in which case the stenting’s role is palliative.

In principle, the role of the non-palliative esophageal stents is *to dilate* esophageal stenosis, and their dimensions can be manufactured to match those of the stenosis. A number of SEMS models are currently available and used to treat benign or cancerous esophageal stenosis. Hubmann and colleagues at Linz General Hospital in Austria were the first to use SEMSs *to compress* esophageal varices in the treatment of bleeding [64]. In their earlier study, three different SEM types were inserted in the region of gastroesophageal junction in 20 patients for whom previous treatments (sclerotherapy, ligation, and balloon tamponade) had failed to stop bleeding. Upon release, the stents compressed the varices, and the bleeding stopped immediately in all patients. Models of the inserted SEMSs included the Choo stent (M.I. Tech Ltd, Seoul, Korea), Boubela-Danis and Danis stents (both manufactured by ELLA-CS s.r.o., Hradec Kralove, Czech Republic). The stenting devices, which include complex delivery systems and the wire stents as such, are composed of a variety of polymers and metals. During the procedure, all these materials come into contact with mucosal surfaces of the upper GI tract. Although the stents were left in place for 2–14 days, no complication attributable to a tissue reaction to the foreign material was recorded. Over the subsequent 30 days, no recurrent bleeding was observed.

The SX-ELLA^®^ Danis stent is an SEMS specifically developed for arresting esophageal acute variceal bleeding, recommended to remain in situ for no longer than 7 days. The wire is made of nitinol, a biocompatible nickel-titanium alloy displaying shape memory effect, covered with a layer of polyurethane. Subsequent studies [65,66,67,68,69,70] confirmed the promising performance of the procedure, but also suggested that additional interventional procedures should be considered after removal of the stent in order to reduce the portal pressure. In a comparative human trial study [71] it was found that the insertion of a SX-ELLA Danis stent is more effective than the balloon tamponade in controlling variceal bleeding. A national multicenter study carried out in Austria confirmed a similar trend in cases of refractory variceal bleeding, but also found that bleeding-related mortality remained high [72]. In all these reports, however, there is no mention of possible biomaterial-related complications. The few mentioned episodes of ulceration or local necrosis were attributed generally to mechanical irritation.

### 2.4. Topical Hemostatic Agents for Variceal Bleeding

Developing hemostatic agents for the management of GI bleeding is not new, and the idea can be traced back to the previous century [73]. Historically, the development of surgical sealants/adhesives overlapped with that of hemostats; thus far, the difference between them is rather imprecise, although there are adhesives or sealants that are not hemostatic, and vice versa. Closer to our time, a number of topical hemostats have become available for the treatment of bleeding that occurs in the upper or lower GI tract. One such agent is an extract of five plants, developed in Turkey and marketed as Ankaferd BloodStopper^®^. The mechanism of action of this product, which contains more than 40 different enzymes and polypeptides [74], is largely unknown. It has been proposed that it induces a rapid aggregation of the blood erythrocytes and also interacts with plasma proteins [75]. However, it is problematic to qualify this injectable product as a biomaterial rather than a drug, but we should acknowledge the reports that Ankaferd^®^ has been successfully applied to arrest variceal bleeding [73,76,77,78].

Currently, there are two hemostatic agents available for endoscopic therapeutic procedures and are constituted of true biomaterials. The agent TC-325, with the brand name Hemospray^®^, has been developed by Cook Medical Inc., Bloomington, IN, USA [73,79]. It is prepared from a granulated natural mineral material known as bentonite clay (an aluminium phyllosilicate consisting mainly of the montmorillonite mineral). It is highly biocompatible, even edible, does not contain organic matter, and it has strong absorbent properties. The hemostat is delivered onto the bleeding surface as a powder through a device powered by pressurized CO_2_ cartridges. The action mechanism is based on the absorption of water from the surrounding tissues by the TC-325 powder, which becomes an adhesive aggregate generating a mechanical barrier over the bleeding site. Further mechanistic aspects of the process are yet to be investigated [79]. Hemospray^®^ was reported to be useful in emergency management of acute variceal bleeding, serving as a bridge towards more definitive endoscopic procedures, and does not require specific endoscopic expertise. The studies so far [80,81,82] showed that this therapy reduced significantly rebleeding within a short period, with no major events or device-related mortalities. Another hemostatic powder is EndoClot^®^ PHS (EndoClot Plus Inc., Santa Clara, CA, USA). According to the manufacturer, this biocompatible product is a modified polysaccharide (from starch) that they call “absorbable modified polymers”, which was specifically developed as a hemostat to treat GI bleeding, and is deliverable by using a pressurized air pump device. The mechanism of the EndoClot-induced hemostasis is based on the water-absorbing properties of the powder that triggers the concentration of blood cells and coagulation proteins and accelerates the clotting process, resulting in the formation of an adherent barrier to further bleeding. A very recent comparative study [83] has shown that both Hemospray^®^ and EndoClot^®^ PHS powders are equally effective as hemostats in the treatment of GI bleeding, with no differences in short- or long-term success and rebleeding events. Thus far, this is the only report regarding the use of EndoClot^®^ PHS for variceal bleeding.

What is remarkable about the two hemostatic agents discussed above, and also different from most of the biomaterials developed for the GI tract, is that their biocompatibility has been assessed prior to marketing by the manufacturers, who also disclosed the chemical composition of these products.

### 2.5. Obturation with Cyanoacrylate Glue

To date, the cyanoacrylate glues are the most successful surgical adhesives available to medical specialists. They have a rather unique position among biomaterials. As such, alkyl-2-cyanoacrylate monomers are highly reactive substances, therefore able to induce aggressive response when applied to tissues, and―at that stage―not considered to be biomaterials. However, upon contact with physiological water and tissue proteins’ amino groups, the monomers become rapidly solid polymers due to a very fast on-site polymerization process. These polymers display adhesive and sealing properties and fulfil the definition of biomaterials.

Alkyl-2-cyanoacrylates (alternatively, α-cyanoacrylates) were patented a long time ago [84] by Goodrich Co., Charlotte, NC, USA, without the realization of their outstanding adhesive properties. It was the Eastman Kodak Co. (Rochester, NY, USA) that launched them as adhesives in industrial and medical markets [85]. There is vast literature on cyanoacrylate polymers and their medical applications, as summarized in some recommended reviews [86,87,88,89,90,91]. Proposed in the late 1950s, the cyanoacrylate glues have been soon employed by surgeons for creating sutureless wound seals, and early pioneering work has been carried out in vascular surgery [92,93], ophthalmic surgery [94,95], and surgery of the GI tract [96,97]. Currently, the range of commercial cyanoacrylate surgical adhesives/sealants is extensive, including as examples the following alkyl-2-cyanoacrylates (nature of alkyl group, commercial names, and manufacturer) as the best known products. *n*-Butyl: Histoacryl^®^ (B. Braun Melsungen, Hessen, Germany), Indermil^®^ (Henkel Loctite, Bridgewater, NJ, USA), Glubran^®^ and Glubran2^®^ (GEM s.r.l., Viareggio, Italy), PeriAcryl^®^ (GluStich, Delta, Canada); isobutyl: Iso-Dent^®^ (Ellman International Inc, Hicksville, NY, USA); 2-octyl: Dermabond^®^ (Ethicon, Blue Ash, OH, USA), SurgiSeal^®^ and FloraSeal^®^ (Adhezion Biomedical LLC, Wyomissing, PA, USA), Octylseal^®^ (Medline, Waukegen, IL, USA), Derma+flex^®^QS (Chemence, Corby, UK); blend *n*-butyl/2-octyl: LiquiBand*^®^* (Advanced Medical Solutions, Winsford, UK); *n*-hexyl: IFABOND^®^ (Péters Surgical, Bobigny, France).

Obturation of esogastric varices with cyanoacrylate glue is an established technique, which has been discussed in major topical reviews [14,15,16,18,89,90,98,99,100]. The very first attempt at using cyanoacrylates against varices was an indirect approach [101]: veins feeding the esophageal varices were occluded with isobutyl 2-cyanoacrylate injected through Teflon^®^ catheters. The outcomes were “much more satisfactory” than those resulting from the use of sclerosants. A more direct approach was applied later when isobutyl or *n*-butyl cyanoacrylates were injected intravariceally or perivariceally [102,103]. In principle, the current procedure consists of an endoscopic intravenous injection of the glue, placed intravariceally. Following the fast hardening of the material inside the vein, a plug is generated which blocks the blood flow and stops the bleeding. The polymer plug is eventually extruded into the varix lumen, which can happen any time within one week to one year, when rebleeding may recur. Dilution of the glue with lipiodol (a mixture of iodized ethyl esters of poppy seed oil fatty acids) is sometimes practiced, aiming at preventing the occlusion of the endoscopic channel, controlling the speed of hardening, reducing the risk of embolization, or functioning as a radio-opaque contrast agent when necessary. However, the administration of undiluted cyanoacrylate glues is preferred. The procedure has been attempted as a treatment for all types of varices, but is mainly applied for the obturation of fundic varices (types GOV2 and IGV1), and is considered to be one of the most effective therapies for variceal bleeding, although there is a large variability in reported comparative rebleeding rates. 

The complications of cyanoacrylate glue application to bleeding varices are mainly of a thrombotic nature due to embolization events caused by the migration of the solidified glue particles or of the thrombus formed on the surface of glue plugs, and include cerebral embolization and stroke, and pulmonary, coronary, splenic, or renal embolization. Other reported complications included inflammation, bacteremia and sepsis, hemoperitoneum, erosion and mucosal edema, and gastric ulceration. The thrombotic complications are obviously related to the presence and migration of dispersed solid material, but this is not specific only to cyanoacrylate glues. Bacteremia and infection are also common complications induced by any foreign material or device that are not properly sterilized, although it has been hypothesized that they are inherent to pathogenesis of variceal bleeding [104]. However, the edematous or ulcerogenic effects may be related to a specific action of the biomaterial, i.e., the glue plug. Perivascular inflammation and vessel wall necrosis have been attributed to a foreign-body reaction [105].

In contrast to most of the biomaterials used in the GI tract, the biocompatibility of cyanoacrylate glues has been extensively investigated right from the time of their introduction in medicine [106,107,108]. Initially, it was suggested that the high reactivity of the monomer and/or the heat released upon its polymerization may be causes for inflammatory response and necrosis, and―to a certain extent―they could be. The unusually high reactivity of 2-cyanoacrylate monomers is attributed to a strong electron-withdrawing effect of the cyano (–C≡N) and alkoxycarbonyl (RO–CO–) groups, both attached to the α-carbon of the polymerizable double bond H_2_C^(β)^=C^(α)^. As a result, the β-carbon can be activated even by weak nucleophilic bases like water or alcohol, rapidly initiating the polymerization process. The pendant amino groups in the blood or tissue proteins may also function as nucleophilic agents able to initiate the polymerization [106]. The shorter the alkyl substituent (R–), the stronger the activation of the double bond, and the faster the polymerization rate. That explains why the cyanoacrylates with longer alkyl chain lengths, such as *n*-butyl, *n*-hexyl, or 2-octyl cyanoacrylates, take more time to polymerize and are easier to apply. These cyanoacrylates also result in a higher flexibility of the glue cast over the varix, induce lower histotoxicity, and are more effective in achieving hemostasis.

Currently, it is believed that there are two potential sources for biomaterial-related complications that are due to the chemical nature of the glue: the low tissue absorption of the glue leading to a foreign-body reaction, and the biodegradation of the solidified polymer and concomitant accumulation of toxic by-products resulting from this process. The glue-tissue foreign-body type reaction is accompanied by the infiltration of polymorphonuclear cells and macrophages that attempt to phagocytize the glue and tissue debris. Generation and accumulation of degradation toxic by-products (mainly formaldehyde and cyanoacetates) cause cell death and the release of oxygenated free radicals leading to formation of lipid peroxides and promotion of inflammatory mediators and thromboxanes. Both mechanisms result eventually in ischemia, necrosis and tissue damage. The toxicity of poly-2-cyanoacrylates has been investigated in vitro (abiotic conditions) [107], in vitro (cell cultures) [106,109,110,111,112], and in vivo (animals) [108,113], to mention just a few landmark reports. The topic has been also episodically reviewed [89,90,114,115]. It is generally accepted that the amount of toxic formaldehyde and other chemicals generated through the biodegradation of poly-2-cyanoacrylates is sufficiently low to allow their complete metabolization within tissues and subsequent clearance by the physiologic flow. The resulting metabolites are eliminated through normal excretory routes, and no material is stored in the tissues.

## 3. Resection Devices for Polypectomy

Endoscopic resection of colorectal pedunculated polyps, known as colonoscopic polypectomy, is an essential tool in preventing colorectal cancer. The post-polypectomy hemorrhage and the incomplete polyp removal are the most serious complications of polypectomy, and efforts are continuously made to reduce their incidence.

The use of snares, known as “endoloops”, is the universally accepted method to perform polypectomies. Tsuneoka and Uchida, who apparently also coined the term “polypectomy”, were the first to use endoloops made of stainless steel wire for the successful resection of polyps within the stomach [116]. In further developments, coagulating and cutting electric currents were passed through the wire to prevent hemorrhage and sever the polyp’s stalk [117]. Following other improvements, success has been reported in larger series of patients [118]. The “double-snare method” has been later developed [119], consisting of two plastic-coated snares, one with the role to strangulate the blood flow and the other, placed above, with the role of cutting the stalk. Nylon^®^ as a material for the endoloops became increasingly popular as the detachable snares have been developed [120], and in 1991, in collaboration with Olympus Optic Co in Tokyo, Hachisu [121] established a model of detachable endoloop that is basically the prototype for all models available on the market. Olympus remains the major manufacturer of endoloops both for polypectomy and for variceal ligation [122].

From a material view, an endoloop consists of an outer polymer sheath (usually Teflon^®^), an inner stainless steel coil, and the loop (made of Nylon^®^) attached to a stopper made of silicone rubber (for adjusting the tightening of the loop). Following the endoscopic procedure, the loop material is the only part of the device that remains in prolonged contact with the tissues. As Nylon^®^ (a polyamide) is a proven biocompatible material, this can explain why there were no evidence of biomaterial-related effects while the loop was still inside the GI tract. Within 4–7 days, the loops slough off spontaneously and are then eliminated in the feces with no side effects. A number of retrospective studies [123,124,125,126,127,128] have confirmed the safety and efficacy of the polypectomies performed with endoloops, with no mention of complications that could have been caused by the material of the device. For improving performance in preventing hemorrhage, the endoloops have been used in association with band ligation [129,130], clipping [131,132,133], and with the recent technique of endoscopic mucosal stripping [134].

## 4. Clipping for Endoscopic Control of Bleeding

Endoscopic clipping using metallic devices known as “hemoclips” or “endoclips” has been developed for achieving hemostasis in a relatively large number of non-variceal bleeding conditions, including: esophageal ulcer; Mallory-Weiss tear syndrome; gastric, duodenal and rectal ulcers; gastric tumours; Dieulafoy’s lesion; post-polypectomy and post-sphincterotomy hemorrhages; and diverticular bleeds. The hemoclips can be applied to active bleeding sources (either sprouting or oozing), to nonbleeding visible vessels and stigmata of hemorrhage, or for closing perforations and fistulae. They were first introduced in practice in Japan [135], and subsequently, their use has widened following certain improvements in both technique and devices [136,137,138]. Injections of epinephrine, ethanol or polidocanol (ethoxylated dodecanol) and/or coagulation (electric or thermal) can accompany the application of hemoclips. They have been also used to assist the placement of endoscopic feeding tubes or the adjustment of bile duct for endoscopic retrograde cholangiopancreatography.

Currently, hemoclipping is considered a safe and effective alternative for controlling hemostasis in non-variceal bleeding episodes within the GI tract [139,140,141,142,143], offering also the great benefit of not affecting the healing processes. There are many hemoclip models on the market, for instance, the following (the brand names may actually be followed by trademark symbols (^®^, ™): EZ clips, hemoclips HX-600/610/200, QuickClip, QuickClip2 (Olympus Medical Corp, Tokyo, Japan or Olympus America Inc, Lombard, IL, USA); TriClip, Instinct (Cook Endoscopy/Cook Medical, Bloomington, IN, USA); Resolution clip (Boston Scientific, Rolling Meadows, IL, USA); MultiClip Applier (InScope/Ethicon, Blue Ash, OH, USA); OTSC (Ovesco, Tubingen, Germany); DuraClip (Conmed, Edison, NJ, USA). The models can be either single-use or reusable devices. In general, the endoscopic hemoclips differ mainly in their mechanisms of deployment, which are outside of the body or in very short contact with internal mucosae. Current developmental work is mainly dedicated to creating the simplest system to release the clip from the delivery catheter. The majority of published reports have been related to the hemoclips manufactured by Olympus Corp, and no comparative studies are available regarding the performance of different devices.

From a biomaterials perspective, in a GI hemoclipping system the parts that penetrate the patient’s body contain metals (e.g., cables, forceps mechanisms) and polymers (e.g., Teflon^®^ sheaths). These parts, however, are subjected to very short direct contact with the tissues or mucosae and no foreign-body response is expected. The clips as such are made of stainless steel, and these are the only components actually left at the site of application after the treatment has finished. It is traditionally believed that a hemoclip spends an average of 2 weeks before sloughing off and then being eliminated naturally from the body. In the absence of any published research dedicated to the retention time, this value is rather speculative: much longer durations have been reported, even in excess of 2 years [144,145,146]. The existing literature indicates that rebleeding is virtually the only complication of hemoclipping, such suggesting that the clip material may be highly biocompatible and does not induce other complications and, therefore, implying that its retention time in the body is of little significance. However, it should be taken into account that the presence of metallic objects in the body can interfere with imaging techniques such as magnetic resonance imaging and computed tomography. To avoid this interference, clips made of polymers (either absorbable or not) have been introduced in laparoscopic surgery, but we are not aware of similar developments for the endoscopic clips. 

## 5. Biomaterials in Endoscopic Bariatric Therapies

The term “bariatrics” designates a branch of medicine dedicated to the study of obesity, including its causes, prevention, and treatment. Obesity has been recognized by the World Health Organization as an epidemic, soon potentially affecting about one third of the world’s population. Consequently, an impressive multitude of strategies have been developed for the medical management of this condition, including changes in lifestyle (diet, physical exercise), pharmacotherapy, bariatric surgery, and endoscopic bariatric therapy (EBT). This section concentrates on EBT from a biomaterials viewpoint.

EBTs were developed to fill the treatment gap in the management of obesity. This gap refers to the patients who could not achieve meaningful weight loss through lifestyle changes or pharmacotherapy, those with moderate obesity who do not qualify for surgery, and those with severe obesity who refuse surgery. The range of EBT procedures is extensive, reflecting how serious the problem of obesity is. In our discussion, we benefit of a number of excellent reviews [147,148,149,150,151,152,153,154,155] covering the vast literature dedicated to the EBT techniques and devices. As compared to other devices routinely inserted in the GI tract, those specific to EBTs should be most exposed to biocompatibility-related effects. The majority of the devices for EBTs are large or very large in size, thus involving substantial volumes of biomaterials that are in intimate contact with the internal mucosae and tissues of the GI tract for much longer durations than other devices. Such situation is expected to elicit foreign-body responses and subsequent side effects.

To classify the great diversity of endoscopic bariatric techniques and devices can be a tedious task. Five categories of EBT procedures/devices are generally recognized, including: space-occupying devices; restriction procedures (plication); bypass liners (involving small bowel); aspiration therapy; and electrical stimulation. Most of these are not approved by the Food and Drug Administration (FDA), but many are in experimental usage. We will discuss here the procedures where endoscopy is used for both placement and removal, or only for one of these stages. The mentioned brand names may actually be followed by trademark symbols (^®^, ™).

In the first category, intragastric balloons (IGBs) are traditionally employed to reduce the gastric space and induce a feeling of satiation. The balloons are usually removed after 6 months, but some of them are allowed longer. The best known models include Orbera (Apollo EndoSurgery, Austin, TX, USA), ReShape Duo (ReShape Medical, San Clemente, CA, USA), Spatz (Spatz Medical, NY, USA), and Obalon (Obalon Therapeutics, Carlsbad, CA, USA); Heliosphere (Helioscopie, Vienne, France); Silimed (Silimed, Rio de Janeiro, Brazil). The material for Obalon has not been disclosed, but the other are all made of silicone elastomers. After placement, they are filled with gases (air, helium, nitrogen, sulfur hexafluoride, or mixtures thereof) or liquids (usually saline). The performance of IGBs is primarily assessed by the weight loss they are inducing in the patients. However, their complications are numerous and diverse, including patient’s intolerance (pain, nausea, reflux), gastric disorders, lesions and obstructions along the GI tract, and gastric ulcer or perforation. Accidental deflation of the balloon or manifestations of intolerance require its immediate removal by endoscopic means. To our knowledge, there are no studies on a possible link between any of these complications and the chemical nature of the biomaterials in balloons. This may be a result of the remarkable inertness of the silicones, in this case supported by a relatively satisfactory clinical performance of the IGBs. Amongst the non-balloon space-occupying devices, the TransPyloric Shuttle (BAROnova, San Carlos, CA, USA) is a two-component device made of silicone, which is placed behind the pyloris to induce satiety through delay in gastric emptying. Another device in this category, the Full Sense (BFKW, Grand Rapids, MI, USA) is a simple combination of a stent connected to a disk, both made of nitinol wires embedded in silicone. It is placed endoscopically within the distal esophagus and cardia, where its presence induces the feeling of fullness. Both devices are still at the stage of clinical trials and there are no data on complications caused after 6 months of retention by poor biocompatibility of materials. 

The plication procedures are endoscopic incisionless techniques able to delay gastric emptying. The Endoscopic Sleeve Gastroplasty (ESG) consists of a sleeve along the curvature of the stomach by using the OverStitch (Apollo Endosurgery, USA) suturing device. Inflammatory response has been reported as a complication, which may be related to a tissue reaction to sutures. Another method, Primary Obesity Surgery Endoluminal (POSE), makes use of an incisionless operating system (USGI Medical, USA) to generate anchor-like plications in order to reduce the gastric volume. Amongst the procedures involving the small bowel, the EndoBarrier (GI Dynamics, USA) is a Teflon^®^-coated nickel-alloy implant (65 cm in length) that is endoscopically deployed into the duodenum extending to the jejunum, to create a barrier between the pancreatic and biliary secretions and the undigested food. The device is designed to stay in situ for 12 months. Because of some adverse effects in clinical trials (e.g., hepatic abscess, pancreatitis, bleeding), improved models have been developed and are currently trialed. Aspiration therapy is another endoscopic procedure where the ingested food is partially removed through a percutaneous gastrostomy tube (A-Tube; Aspire Bariatrics, USA), which is made of silicone. It is recommended to be maintained no longer than 5 years. The most frequent complications comprised formation of stoma granulation tissue and stoma infection.

To conclude, the bariatric applications involve the presence in the GI tract of large amounts of biomaterials that are maintained in intimate contact with mucosae and tissues for long periods of time. The reported adverse effects have not been linked directly with biocompatibility-related effects. However, there has been no research so far addressing specifically this issue.

## 6. Endoscopic Anti-Reflux Devices

Gastroesophageal reflux disease (GERD) is a chronic condition that can be caused by defects in the lower esophageal sphincter (LES) and affects a large segment of population. The treatments include lifestyle changes, pharmacotherapy, laparoscopic surgery and endoscopic anti-reflux devices. As only a few of the patients with GERD accept surgery, there was a significant progress over the past two decades in the development of endoscopic procedures and devices to treat in a less invasive manner this condition [156,157,158]. In our following discussion, the mentioned brand names may actually be followed by trademark symbols (^®^, ™).

The anti-reflux procedures/devices employed both in the past and currently can be categorized into three groups: (a) techniques to generate fundoplication; (b) electrothermal treatment of the tissues at the LES level; (c) injection/implantation of foreign material into LES for bulking effects. Due to lack of adequate performance and to unacceptable adverse effects, most of these devices have been taken off the market, including EndoCinch (CR Bard, Tempe, AZ, USA), Wilson-Cook Endoscopic Suturing Device, and NDO Plicator (NDO Surgical, Mansfields, MA, USA) in category (a), as well as Enteryx (Boston Scientific, Rolling Meadows, IL, USA), Gatekeeper Reflux system (Medtronic, Memphis, TN, USA) and injection of Perspex microspheres in category (c). Therefore to discuss them is rather superfluous. Currently, as indicated by ASGE [158], there are only three procedures in use that are approved by the FDA. Of these, the Stretta system (Mederi Therapeutics, Greenwich, CT, USA), in category (b), for thermal treatment of LES tissue with radiofrequency in order to induce fibrosis in the submucosa and muscles, does not involve biomaterials and will not be discussed here. The other two systems, both belonging to category (a), i.e., transoral incisionless fundoplication with the use of EsophyX device (Endogastric Solutions, Redmond, WA, USA), and Medigus Ultrasonic Surgical Endostapler (MUSE; Medigus Ltd, Omer, Israel), are sophisticated engineered devices that can be handled endoscopically and are able to modify the angle between the cardia and the esophagus by generating esophagogastric plication. While the EsophyX device achieves the aim by fasteners (SerosaFuse; Endogastric Solutions, Redmond, WA, USA) made of polypropylene (PP), the MUSE device is using staples made of titanium. Both the PP fasteners and titanium staples are supposed to stay indefinitely in place. As the biocompatibility of these materials is well documented, related complications are not expected. The adverse effects reported [157,158] have not been attributed to biocompatibility issues. 

## 7. Biliary and Pancreatic Stents

Stenting is used in the treatment of esophageal stenosis. Benign stenosis can be caused by reflux esophagitis or as a result of surgery for anastomosis. Cancerous stenosis is a result of esogastric or other tumours. Although preferred in esophageal applications, the stents made of polymers are progressively replaced in other applications by metallic stents able to expand, the SEMSs.

Currently, there is a large variety of SEMSs commercially available to the gastroenterologists as esophageal, biliary, gastroduodenal, pancreatic and colorectal stents [59,60,61,62,63,159,160,161,162,163] to be used in the treatment of strictures or obstructions of benign or cancerous origins. The metallic parts are made of stainless steel or alloys. Nitinol, a frequently used material, is a nickel-titanium alloy that displays shape memory effect and high biocompatibility, and is in demand as a biomaterial (catheters, stents, wires, filaments, needles, intrauterine devices). When a nitinol wire has a platinum core, the material is known as “platinol” (a term not to be confounded with the anti-cancer drug). Another metal used in SEMSs is the nickel-cobalt-chromium alloy, known as “elgiloy”. Commonly used SEMSs include Wallstent^®^ and Ultrafex^®^ (Boston Scientific, Rolling Meadows, IL, USA), Evolution^®^ (Cook Medical, Bloomington, IN, USA), Z-Stent^®^ (Wilson-Cook, Winston-Salem, NC, USA). The wires in stents can be: (a) covered (coated) with layers of polymers such as silicones, Teflon^®^ and other fluorinated polyolefins, polyurethanes, or poly(ε-caprolactone); (b) uncovered; or (c) partially covered. The polymeric coat increases the tendency of stents to migrate.

Stents made of synthetic polymers are still used extensively. In a recent review [163], a large variety of plastic stents used in biliary and pancreatic applications is described, as marketed by well-known suppliers such as Boston Scientific, USA; Cook Endoscopy, USA; or Olympus, Japan. The polymers involved include polyethylene, polyurethanes, polyethylene/polyurethane blends, EVA copolymer, or Teflon^®^ and other fluorinated polyolefins, all these being materials with well documented records of biocompatibility in the human body. The plastic stents have the drawback of getting occluded due to bacterial biofilm growth and biliary sludge accumulation [164], sometimes in a shorter time than the survival period of the cancer patients. While SEMSs pushed the service life of the stents to 6 months or longer, they also have drawbacks, for instance getting embedded into the duct walls. Covering the metal wire with polymers solved this issue, but increased the risk of stent migration, albeit the covered SEMSs may offer generally a lower risk of dysfunction [165] in spite of previous different opinions [166]. 

As seen frequently in our present analysis of other GI devices, there is not much insight into possible adverse effects caused by lack of biocompatibility of the stent materials. Indeed, no complications have been attributed to such potential factor. Perhaps the best proof for the satisfactory biocompatibility of the stent materials was provided by the study of “forgotten” stents [167], where biliary stents had to be removed surgically or endoscopically after 2–10 years since their placement, because of sudden complications, such as pain, cholangitis, jaundice, and stones. No definite association between the complications and the stent materials could be inferred.

## 8. Conclusions

Regarding the nature of biomaterials of which the therapeutic devices for gastrointestinal endoscopy are made, there appears to be―with a few exceptions―a minimal exchange of disclosed information between the medical specialists using them and their manufacturers. It is possible that the lack of doctors’ interest in learning about chemical nature of the routinely used devices or implants might have prompted the manufacturers to disregard a need for providing details about the materials involved in making their products since, anyway, the applications of these devices in the GI tract appeared to be successful frequently in treating various conditions. Nevertheless, we cannot overlook that this situation may also be convenient to the manufacturers in keeping some control on the market competition. This state of affairs might have two consequences. First, regarding the multitude of complications reported in the gastroenterological literature it seems that they have been rarely correlated with a possibly poor biocompatibility of the materials in the endoscopic devices, even when some of the complications could have been caused by it. Second, based on such incomplete knowledge, one would envisage little progress in the development of an in-character subdiscipline of gastroenterologic biomaterials. However, our review also shows that developmental work and clinical trials involving a significant range of endoscopic devices and techniques are continuously pursued, leading to various levels of device improvements and clinical successes, also suggesting applications with greater clinical potential in the future. This will certainly contribute to the establishment of such a subdiscipline.

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
