# Peer review of "Biomaterials in Gastroenterology: A Critical Overview"

_medicina, 2019, doi:10.3390/medicina55110734_

Round 1

Reviewer 1 Report

Nice and extensive review.

1- Unfortunately there is no mention of the process of allowing all these devices to be used. to my knowledge these devices need FDA approval in the USA. i am sure similar regulatory pathways are followed in Europe and New Zeland. it would be nice to have a review of the standard used to allow these devices to be used extensively. the bio compatibility issue would arise because of exposure of the mucosa to all these devices. some have more than one bio material component .   missing is the degree of bio degradation of the biomaterial upon exposure to different secretions in the gut.

2- Also missing is the bio compatibility that needs to be addressed with the advent of cell therapies and gene therapy. All of these procedures require the biopsy of cells and have the cells exposed to different biomaterials before re-introduction into the patient. This is  the future of medicine .

Author Response

Ms: medicina-615021

Response to Reviewer #1.

/1.1/  The entire information that was available about the FDA-related status of the materials/devices reviewed in the manuscript has been mentioned wherever appropriate. The Reviewer should consider that many of the devices discussed here have been already discontinued regardless of their FDA status. The field of the regulatory bodies’ (FDA, Euro etc.) decisions regarding the use of medical devices is quite dynamic, with devices that had their approval revoked, or are marketed without approval, or have re-gained the FDA approval after an interruption, or are in course of an appeal to a regulatory decision, to the extent that is difficult to keep an accurate track of such events. This manuscript is a scientific review, not a marketing report, therefore if the reviewer wished to get more information on the current regulatory status of the materials/devices discussed in our overview, there is the chance of carrying out an online search.

/1.2/  We do not understand the Reviewer’s statement, “it would be nice to have a review of the standard used to allow these devices to be used extensively”. Well, obviously our review is not about the particular aspect of test standards (if we understand correctly the statement), therefore nothing can be done. Does the Reviewer suggest that we should add a review about test standards? What about maintaining the topic as intended, and about the space allowed by the journal?

/1.3/  We believe that the biocompatibility issues related to the GI tract and involving mucosal structures have been extensively discussed in the Introduction, paragraphs 1.1 and 1.2. Whatever was published about the degradative processes affecting the materials exposed to gut secretions has been included in our manuscript. What information would this Reviewer wish to be included additionally? We will appreciate a clear indication. We should also mention that a leading idea of this overview was to show the disproportion between the level gastroenterology has reached as a medical branch and the minor volume of research dedicated to the materials employed in the devices. If there is nothing more published on degradation of materials in the GI tract, what the Reviewer would suggest that we should do?

/2/  Our manuscript did not have as an aim to discuss cell or gene therapies. The subject is already vast enough that a separate treatment of the topics would be necessary in separate reviews. With all due respect, we have to remind to this Reviewer that our topic is BIOMATERIALS, not the further developmental stages illustrated in tissue engineering, regenerative medicine, cell therapy, or gene therapy. The Reviewer should bear in mind that there is a page limit imposed by the journal, and the limit would have been surpassed by tens of pages and hundreds of references if we would have also discussed the mentioned therapies.  Yes, we know that “this is the future of medicine”, but we have focused here on the topic of BIOMATERIALS. We hope the Reviewer will eventually understand the issue.

For an improved understanding of the topic of the overview, and as a response to the Reviewer’s criticism, we have re-structured the manuscript to an extent that hopefully will satisfy the Reviewer.

Reviewer 2 Report

1- Author should focus on introduction, at present there are no correlation between paragraphs and the rationale of the review. The non-expert on the topic will have a difficult time following this introduction and understanding its relevance.

2- Authors should include figures in each section to make this review more promising, insert at least 8-10 figures.

3- Authors can improve conclusion which can highlight major finding and their future perspective.

4-All the references are old, include some recent references and specifically  insert some related recent references from"Medicina".

Author Response

Ms: medicina-615021

Response to Reviewer #2.

/1/  The reviewer is worried that a “non-expert” will have problems to understand the topic as exposed in our Introduction. We surmise the Reviewer refers to a non-expert in biomaterials. It is precisely what we have done in the Introduction: make easier for gastroenterologists to understand the biomaterials issue. We have defined and described the concepts of biomaterials, biocompatibility, biomaterials interactions with the GI tract, and summarized elements of tissue engineering and regenerative medicine. All this was aimed at providing understanding knowledge to the non-expert. Unfortunately this was not enough for the Reviewer, who was not able perceive any “correlation between paragraphs and the rationale of the review”, a rather strange statement considering that the same reviewer gave maximum score for the significant contribution of our manuscript to the field. It appears that the Reviewer had himself/herself problems with understanding the topic of the manuscript, but we cannot do much to alleviate such predicament. All terms dealt with in this article have been clearly defined and extensively commented in the Introduction. Does the Reviewer really suggest that we should write another Introduction, or add more text to it, only because he/she does not understand some matter? Within 10 days allowed for revision? With all due respect, this is unnecessary and practically impossible.

/2/  This manuscript was intended an overview that does not need figures. The addition of 8-10 figures as suggested by the Reviewer will double the volume of the paper, such surpassing by far the limit allowed by the journal. Within only a few days left until the deadline for re-submission, does the Reviewer seriously suggest that we should made up illustrative material, or select figures from other publications requiring copyright approval from publishers? This is another unconstructive and unrealistic recommendation that regretfully we cannot take into consideration.

/3/  Yes, it would be nice to be able to comment on major findings and a rosy future for a potential gastroenterologic biomaterials field, but unfortunately there is not much in terms of publications. Continual improvements in the quality of existing devices and innovations are indeed happening in the R&D departments of the manufacturers, however these data are commonly published in glossy brochures, rather than scientific literature. To add empty words to the Conclusions will not smooth the current critical aspects in the development of a discipline of gastroenterologic biomaterials, and we prefer to leave the text as it is.  As authors, we have the right to express our opinion, whether  optimistic or not, and whether the Reviewers like it or not.

/4/  The Reviewer concluded that “all the references are old”. Let us make an estimation: from a total of 167 references that we have cited, a number of 62 (almost 40%) are from year 2010 onwards. Therefore we cannot take into consideration this Reviewer’s incorrect comment. As for Reviewer’s request to add references published in Medicina, all we can say is that we have cited an article published in Medicina, more precisely ref. [100]. We would have used more if the need would have occurred. We question the ethical standard of the Reviewer’s request.

For an improved understanding of the topic of the overview, and as a response to the Reviewer’s criticism, we have re-structured the manuscript to an extent that hopefully will satisfy the Reviewer.
